# Modern Kidney-Sparing Management of Upper Tract Urothelial Carcinoma

**DOI:** 10.3390/cancers15184495

**Published:** 2023-09-10

**Authors:** Alireza Ghoreifi, Reza Sari Motlagh, Gerhard Fuchs

**Affiliations:** 1Institute of Urology, University of Southern California, Los Angeles, CA 90089, USA; alireza.ghoreifi@med.usc.edu; 2Urology Department, University of Vienna, 1090 Vienna, Austria; reza.sarimotlagh@meduniwien.ac.at

**Keywords:** kidney-sparing surgery, radical nephroureterectomy, upper tract urothelial cell carcinoma, ureteroscopy

## Abstract

**Simple Summary:**

Upper tract urothelial carcinoma (UTUC) is a lethal cancer of the urinary tract. Radical nephroureterectomy with bladder cuff excision is the gold standard for the management of patients with UTUC. Nevertheless, less-invasive surgeries to preserve maximum kidney function, such as endoscopic ablation and segmental ureteral resection, have become the preferred options in select patients. In this paper, we reviewed the latest evidence on the kidney-sparing management of upper tract urothelial carcinoma. We showed that these approaches are acceptable for low- as well as select high-risk patients who are not eligible for radical treatments. The main advantages of such approaches include renal function preservation as well as decreased surgical morbidity associated with radical treatments.

**Abstract:**

Purpose: To review the latest evidence on the modern techniques and outcomes of kidney-sparing surgeries (KSS) in patients with upper tract urothelial carcinoma (UTUC). Methods: A comprehensive literature search on the study topic was conducted before 30 April 2023 using electronic databases including PubMed, MEDLINE, and EMBASE. A narrative overview of the literature was then provided based on the extracted data and a qualitative synthesis of the findings. Results: KSS is recommended for low- as well as select high-risk UTUCs who are not eligible for radical treatments. Endoscopic ablation is a KSS option that is associated with similar oncological outcomes compared with radical treatments while preserving renal function in well-selected patients. The other option in this setting is distal ureterectomy, which has the advantage of providing a definitive pathological stage and grade. Data from retrospective studies support the superiority of this approach over radical treatment with similar oncological outcomes, albeit in select cases. Novel chemoablation agents have also been studied in the past few years, of which mitomycin gel has received FDA approval for use in low-risk UTUCs. Conclusion: KSSs are acceptable approaches for patients with low- and select high-risk UTUCs, which preserve renal function without compromising the oncological outcomes.

## 1. Introduction

Upper tract urothelial carcinoma (UTUC) is an uncommon type of cancer with an estimated annual incidence of 1–2 cases per 100,000 [1,2]. The most common risk factors of UTUC in western countries are tobacco and aromatic amines exposure; however, 10–20% of cases are familial and can be linked to hereditary non-polyposis colorectal cancer spectrum disease (Lynch syndrome) [1]. Despite surgery with curative intent, the 5-year cancer-specific survival of UTUC is <50% for stage 2–3 and <10% for stage 4. Moreover, recurrence in the bladder and contralateral upper tract occurs in 22–47% and 2–6% of UTUC patients, respectively [1,3,4].

The incidence of UTUC has been increasing in the past few decades most likely due to improved diagnostic techniques, such as high-quality imaging and flexible ureteroscopy (URS) [2]. This also has led to an increased rate of diagnosis among older patients with an increasing need for less-invasive treatment approaches to preserve maximum renal function [1,2]. Radical nephroureterectomy (RNU) with bladder cuff excision is the gold standard for the management of UTUC regardless of tumor location [3,4]. Nevertheless, kidney-sparing techniques, including endoscopic ablation and segmental ureterectomy (SU), have become the preferred options in select patients, especially those with low-risk diseases [3,4]. Several studies have confirmed the efficacy of these approaches with comparable oncologic outcomes to radical treatments [5]. Nevertheless, recent development in surgical technologies, such as advanced robotic systems and modern ureteroscopes, as well as new ablative modalities, warrant re-reviewing this important topic.

The aim of this study is to review the latest evidence on the modern techniques and outcomes of kidney-sparing management in patients with UTUC.

## 2. Materials and Methods

The scientific paper offers a narrative review of the literature on modern kidney-sparing management strategies for UTUC. The authors conducted a comprehensive literature search on the studies published before 30 April 2023 using electronic databases including PubMed, MEDLINE, and EMBASE. We utilized specific keywords and Medical Subject Headings (MeSH) terms related to UTUC, kidney-sparing techniques, endoscopic treatments, and renal preservation to refine the search and retrieve relevant articles. Additionally, we included reference lists of identified articles for additional sources. Studies were selected based on the English language preference and their relevance to kidney-sparing management of UTUC, specifically focusing on various techniques such as ureteroscopic management, percutaneous approaches, segmental ureteral resection, and novel endoscopic technologies. The literature search identified 203 unique references. We excluded review articles, letters, editorials, and case reports as well as any study that was not relevant as described above. Consequently, 28 studies were included for qualitative synthesis according to the research topic, our inclusion criteria, and data availability. Data extraction involved retrieving important information from the selected studies. Based on the extracted data and a qualitative synthesis of the findings, we provide a narrative overview of the literature. We present the evidence coherently, highlighting the strengths and limitations of the reviewed studies.

## 3. Indications

Pretreatment staging in UTUC patients is challenging due to the limitations of currently available diagnostic tools [1]. The findings of URS/biopsy (tumor grade, focality, and shape), imaging (invasion, obstruction, and nodal status), as well as urine cytology will help in preoperative risk stratification to low vs. high risk for invasive disease (i.e., ≥pT2) [3,4]. Considering these factors, various nomograms and models have been proposed to predict low-risk disease and help with the optimal selection of patients for kidney-sparing surgery (KSS) [6,7,8,9,10]. Based on these data, the European Association of Urology (EAU) and American Urological Association (AUA) expert panels on UTUC proposed two models for pretreatment risk stratification of UTUC to support clinical decision-making (Figure 1). The new AUA guidelines also sub-stratify the patients into favorable and unfavorable to further facilitate risk-adapted management [3].

Current guidelines recommend KSS as a primary treatment option in patients with low-risk UTUC as well as select high-risk cases who have low-volume tumors or imperative indications precluding RNU (e.g., renal insufficiency, single kidney, or bilateral tumors) [3,4]. Taken together, patients who are considered for KSS should preferably have the following criteria: unifocal small-size papillary lesion, negative urine cytology, low-grade ureteroscopic biopsy, and absence of hydronephrosis or invasion in CT imaging [3,4]. In addition, technical feasibility of maximal tumor extirpation and patient compliance with a close follow-up schedule should be considered [11].

## 4. Endoscopic Ablation

Endoscopic ablation, as a KSS option in patients with UTUC, has gained popularity in the past two decades due to the evolution in technology with smaller deflecting endoscopes, advanced lasers, special instruments, and high-quality optics [12].

### 4.1. Techniques

Endoscopic ablation of a UTUC lesion can be performed via a retrograde or antegrade approach. Retrograde is the most common approach; however, the percutaneous method is preferred for larger tumors (>1.5 cm) and those that are difficult to access through a retrograde fashion (i.e., lower pole calyx lesion or prior urinary diversion) [3,4]. The retrograde approach is performed using a rigid or flexible ureteroscope. Using a ureteral access sheath can help with repeated scope passage and also decrease the rate of intravesical recurrence following ablation [13]. On the other hand, the antegrade approach requires establishment of a nephrostomy tract in the correct position [14]. Despite the promising oncologic results for this approach, there is still a lack of evidence regarding its safety profile [15].

The ablation techniques include bulk excision (using biopsy forceps or basket), resection of the tumor to its base (using ureteroscopic resectoscope), and ablation with electrocautery (e.g., Bugbee) or laser energy sources, including thulium (Tm)–yttrium aluminum garnet (YAG), holmium (Ho)–YAG, and neodymium (Nd)–YAG [16]. Ho–YAG is characterized by a longer wavelength and approximately 0.3–0.4 mm tissue penetration, which makes it suitable for use in superficial ureteral tumors. Nd–YAG has a deeper tissue penetration of up to 10 mm, which is a good option for bulkier tumors. However, its use in the ureter is limited due to the low safety margin that may increase its associated complications. Tm–YAG has gained more acceptance in this setting compared with other types of lasers due to the good coagulation and hemostasis features while having a short penetration depth of about 0.1–0.2 mm [17,18,19,20]. A recent systematic review on the use of Thulium lasers in UTUC reported no intraoperative complication and 10.5 to 38% rate of postoperative complications, most of which were mild and transient [21].

Novel endoscopic techniques, such as en bloc enucleation, have also been reported in the literature [22,23]. Although this approach was shown to be feasible in select cases with the advantage of improved histopathologic information, its indications and oncological safety have yet to be determined.

### 4.2. Adjuvant Instillation

Older studies on the use of adjuvant upper urinary tract instillation of BCG or mitomycin C following endoscopic ablation of UTUC have shown comparable results to unrented patients [24]. However, recent studies have demonstrated promising oncological outcomes in these patients. Gallioli et al. reported 52 UTUC patients treated by endoscopic ablation, of whom 26 received immediate adjuvant single-dose upper urinary tract instillation of mitomycin. On Cox regression, mitomycin instillation was associated with a 7.7-fold lower risk of urothelial recurrence [25]. In addition, Labbate et al. recently reported a 63% ipsilateral disease-free rate at 6.8 months following endoscopic ablation and adjuvant mitomycin gel instillation [26]. It is noteworthy that all available studies suffer from small sample size and lack of control groups. In addition, the rate of ureteral stenosis has been reported to be as high as 19% in recent series of adjuvant mitomycin gel instillation [24]. Therefore, the guidelines suggest adjuvant pelvicalyceal chemotherapy instillation following UTUC ablation, albeit as an optional part of routine practice, provided that there is no perforation in the urinary system [3,4].

### 4.3. Follow-Up

There is no high-level evidence regarding the optimal follow-up schedule in patients undergoing endoscopic ablation, and the recommendations are mostly based on experts’ opinions. Current guidelines recommend repeat URS within three months following initial ablation to check for residual disease and/or recurrence [3,4]. In a study of 41 patients who underwent second-look URS, 6–8 weeks following endoscopic ablation for UTUC, cancer was detected in more than half of the patients, of whom 86% were in the same location as the first URS [27]. These findings underscore the importance of second-look URS following initial ablation. Surveillance URS should then be then continued every 3–6 months until no evidence of upper tract disease is identified (preferably up to 5 years). The surveillance intervals depend on tumor grade (low vs. high) and the indication of KSS (imperative vs. non-imperative); patients with high-grade UTUC and those with imperative indications will require closer follow-ups. In addition, CT urogram, cystoscopy, and urine cytology should be included in the follow-up workups [3,4].

### 4.4. Outcomes

The main goal of endoscopic ablation for UTUC is preserving renal function without compromising the oncological outcomes. There is no prospective study comparing endoscopic management with RNU for UTUC. However, the available data from retrospective studies have shown similar oncological outcomes between these two treatment modalities (Table 1 and Table 2) [28,29,30,31,32,33,34,35,36,37,38]. In a recent systematic review and meta-analysis, including 13 studies, Kawada et al. reported that endoscopic management compared with RNU was associated with similar overall survival (OS) (Hazard Ratio: HR 1.27, 95% CI 0.75–2.16), cancer-specific survival (CSS) (HR 1.37, 95% CI 0.99–1.91), and bladder recurrence-free survival (BRFS) (HR 0.98, 95% CI 0.61–1.55). However, the results of this systematic review should be interpreted with caution given the retrospective nature of included studies as well as selection bias due to the heterogeneity of patient populations and inclusion criteria [39].

Despite favorable oncological outcomes of endoscopic ablation, approximately 20–30% of patients may develop disease progression requiring salvage RNU [40]. In a study with a large sample size of 279 patients undergoing endoscopic management for UTUC, Chen et al. reported a 24% rate of salvage RNU. The authors showed that among patients with recurrence following endoscopic ablation, those undergoing salvage RNU compared with others had a better disease-free survival rate (92% vs. 77.5%) as well as a lower rate of UTUC-related death (7.8% vs. 22.5%) [41].

Endoscopic ablation is associated with a better or similar postoperative kidney function compared with RNU [30,33,34,42,43] (Table 3). In a study comparing 20 patients undergoing endoscopic ablation compared with 178 RNU cases, Fejkovic et al. reported better postoperative kidney function in the ablation group [30]. On the other hand, in a study comparing 84 cases of endoscopic ablation and 272 patients undergoing RNU, Chen et al. reported no significant difference in postoperative renal function, chronic kidney disease, or end-stage renal disease [33]. It is worth mentioning that all these studies are retrospective and their outcomes are affected by selection bias and short-term follow-ups.

## 5. Segmental Ureterectomy

Although the feasibility of proximal and total ureterectomy has been shown in the literature [44,45], distal ureterectomy followed by ureteroneocystostomy ± psoas hitch/Boari flap forms the most common type of segmental resection in UTUC patients. It is indicated in low- as well as select high-risk UTUC tumors confined to the distal ureter [3,4]. The main advantage of this procedure over endoscopic ablation is that it provides a definitive pathological stage and grade while preserving ipsilateral renal function.

### 5.1. Technical Considerations

A distal ureterectomy can be performed through open, laparoscopic, and robotic approaches [46,47]. The robotic approach has gained more acceptance in recent years due to favorable perioperative outcomes while ensuring oncologic efficacy. In a study of 15 cases who underwent robotic SU, Campi et al. reported no intraoperative complications and no need for open conversion. Within a 30-day follow-up, 13% of patients experienced grade 3a, yet no ≥ grade 3b, Clavien complications [47].

Similar to RNU, a formal bladder cuff excision with watertight bladder closure is necessary during SU [48,49]. The absence of residual tumor should be confirmed by a negative frozen margin intraoperatively. Lymph node dissection is mandatory in high-risk yet optional in low-risk patients [3,4]. The appropriate template to yield maximal oncologic outcomes remains to be determined; however, dissection of the ipsilateral obturator and external iliac as well as (preferably) common and internal iliac lymph nodes is recommended in patients undergoing distal ureterectomy [50,51].

### 5.2. Outcomes

There is no randomized clinical trial comparing the outcomes of SU vs. RNU. Current data are based on retrospective studies with a high risk of selection, performance, and detection bias. There are two systematic reviews available comparing the outcomes of SU vs. RNU. The first includes 11 retrospective studies with 3963 patients (SU = 983 and RNU = 2980). The meta-analysis of adjusted data demonstrated similar CSS (HR = 0.90, *p* = 0.47), RFS (HR 1.06, *p* = 0.72), and BRFS (HR 1.35, *p* = 0.39) between the two groups [52]. A second systematic review and meta-analysis was recently performed by Veccia et al., which included 18 studies (all retrospective) comparing 1313 and 3484 patients undergoing SU vs. RNU, respectively. The authors showed no statistically significant difference between the two groups in terms of overall and bladder recurrences, metastases, and cancer-related death. On survival analyses, the SU group showed lower 5-year RFS but similar 5-year MFS and CSS compared with RNU [53]. Finally, a recent study of the national cancer database population, including 9016 RNU and 4045 SU cases, confirmed that long-term survival of SU is not inferior to RNU. In this study, female gender, advanced clinical T stage (cT4), and high-grade tumor were associated with a decreased likelihood of receiving SU, while age > 79 years was associated with an increased probability of undergoing SU [54].

In terms of renal function, available data support the superiority of SU over RNU. Feng et al., in a meta-analysis of the weighted mean changes in peri-operative estimated glomerular filtration rate (eGFR), reported a significant decrease of 9.32 mL/1.73 m^2^ in patients undergoing RNU vs. SU [52]. Similarly, in their meta-analysis, Veccia et al. reported higher postoperative eGFR in patients receiving RNU compared with the SU group [53]. Although these findings are in favor of renal function preservation in patients undergoing SU, the results should be interpreted with caution due to the heterogeneity of cohorts and the effect of possible confounding factors, such as neoadjuvant and adjuvant systemic therapy.

## 6. Novel Chemoablation Therapies and Ongoing Trials

Bacillus Calmette–Guérin (BCG) and mitomycin C have been previously investigated for intracavitary management of UTUC, albeit mainly in the adjuvant setting following endoscopic ablation [55]. Nevertheless, the US Food and Drug Administration (FDA) recently approved mitomycin gel/UGN-101 (JELMYTO, UroGen Pharma) as a first-line treatment for patients with low-grade UTUC [56]. UGN-101 is a water-soluble mitomycin gel with reverse thermal properties that allow for local administration as a liquid with subsequent conversion to a semi-solid gel following instillation into the upper tract. The FDA approval was based on the results of the OLYMPUS trial, which was a phase III, open-label, multicenter study of patients with treatment-naïve or recurrent low-grade UTUC [57]. A total of 71 patients enrolled in this trial and received 6 weekly courses of mitomycin gel followed by URS evaluation. Complete response (primary endpoint, defined as negative endoscopic examination and cytology) was achieved in 58% of the patients, of whom 82% had a durable response in one year (secondary endpoint) [58]. While UGN-101 is approved for low-grade non-invasive UTUCs, a recent study showed promising results in patients with imperative indications, including those with high-grade disease. In this subgroup of high-grade UTUCs, 45% had no evidence of disease at the initial postinduction evaluation [59]. Long-term follow-up is needed to confirm the efficacy of UGN-101 in high-risk UTUC cases.

Ureteral stenosis was the most common treatment-associated adverse event in the OLYMPUS trial and was seen in 31/71 (44%) patients, of whom 6 (8%) required intervention (Clavien grade 3 complication). This was thought to be due to the retrograde approach for mitomycin gel instillation [57]. Using the antegrade approach, Rosen et al. reported a case series of patients receiving mitomycin gel. The authors reported similar oncological outcomes compared with the OLYMPUS trial, yet with a much lower rate of ureteral stricture (1/8 asymptomatic stricture) [60]. These findings were confirmed in a larger retrospective multicenter study of 132 patients who were treated with UGN-101 for low-grade UTUC via a retrograde vs. antegrade approach. In this study, complete response was achieved in 48% of retrograde and 60% of antegrade renal units (*p* = 0.1), while Clavien grade 3 ureteral strictures occurred in 32% of retrograde vs. 12% of antegrade cases (*p* < 0.001) [61].

The most novel modality for the ablation of UTUC lesions is photodynamic agents, which have been used in a phase I trial of WST-11/TOOKAD-Soluble for UTUC ablation. This was an open-label trial using padeliporfin to ablate UTUC lesions. This is a new investigational short-acting photodynamic agent, which produces a novel form of vascular-targeted photodynamic treatment. The results were promising with a 94% overall response within 30 days and a final complete pathologic response of 68%. The most common adverse events following padeliporfin administration were transient flank pain (79%) and hematuria (84%), with no ureteral strictures during follow-up [62].

Based on the results of phase I WSAT-11 trial, the multicenter Phase III ENdoluminal LIGHT ActivatED Treatment of UTUC (ENLIGHTED) trial (UCM301) has been initiated [63]. This is a single arm, non-randomized trial, including new or recurrent low-grade, non-invasive UTUCs. Patients receive 1–3 padeliporfin (vascular-targeted photodynamic) VTP treatments every 4 weeks as an induction therapy followed by repeated maintenance treatments for patients who show evidence of tumor recurrence that is deemed treatable. Primary outcome is the number of patients with complete response, defined as an absence of visual tumor on endoscopy, no evidence of tumor on biopsy (if feasible), and negative urinary cytology by instrumented collection. Secondary endpoints included the duration of response at the entire ipsilateral kidney as well as treatment area at 3, 6, 9, and 12 months postprimary response evaluation; overall renal function at 6 and 12 months; development of ureteral obstruction and/or ureteral stent placement; and duration of response/renal function on long-term follow-up. This trial is now in the recruiting phase, with an estimated enrollment of 100 participants.

## 7. Conclusions and Future Directions

KSSs, including endoscopic ablation and segmental ureterectomy, are acceptable approaches for patients with low-risk UTUC as well as select high-risk cases who are not eligible for radical treatments. The only level I evidence in this setting is the use of mitomycin gel in low-risk UTUCs. The feasibility and safety of other types of KSSs have been confirmed in several retrospective comparative studies. The main advantages of KSS include renal function preservation as well as decreased surgical morbidity associated with radical treatments. The key step in KSS is appropriate patient selection, which highly relies on preoperative risk stratification to find low-risk cases. Novel diagnostic and prognostic tools, such as urine-based methylation and blood-based liquid biopsy biomarkers, can help in optimizing preoperative risk stratification and proper patient selection for KSS [64,65]. In addition, these novel markers can be beneficial in the surveillance setting of patients with UTUC undergoing KSS to avoid unnecessary procedures (e.g., URS). On the other hand, the advent of new technologies, such as digital flexible ureteroscopes, as well as novel therapeutic agents, including mitomycin gel and photodynamic agents, may offer more-effective and less-invasive patient care. The efficacy of mitomycin gel was confirmed in a phase III trial that led to FDA approval as a first-line treatment for low-grade UTUC. In addition, the use of padeliporfin, a photodynamic agent, has shown promising results in a phase I trial; however, the phase III trial of this study is still ongoing. While the current data are mainly derived from retrospective studies, ongoing trials are eagerly awaited to shed light on this important topic.

## Figures and Tables

**Figure 1 cancers-15-04495-f001:**
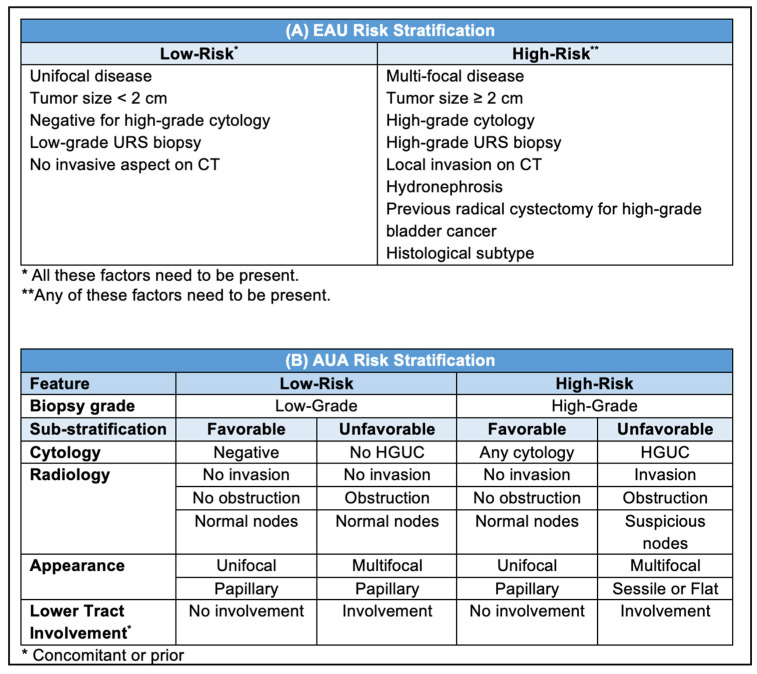
(**A**) EAU and (**B**) AUA pretreatment risk stratification of non-metastatic UTUC. CT: computed tomography; URS: ureteroscopy; HGUC: high-grade urothelial carcinoma [3,4].

**Table 1 cancers-15-04495-t001:** Oncological outcomes of contemporary studies comparing endoscopic ablation vs. RNU for UTUC.

Study (yr) [Ref]	Patients (*n*)		Bladder Recurrence (%)		5 yr OS (%)			5 yr CSS (%)		
	EA	RNU	EA	RNU	EA	RNU	*p* Value	EA	RNU	*p* Value
Lucas et al. (2008) [28]	39	77	5	8	62	72	0.36	82	83	0.98
Cutress et al. (2012) [29]	59	70	42	33	64	75	0.02	85	92.1	0.21
Fajkovic et al. (2012) [30]	20	178	15	36	45	76	0.001	67	91	0.36
Seisen et al. (2016) [31]	42	128	NA	NA	74	73	0.06	83	87	0.18
Vemana et al. (2016) [32]	151	302	NA	NA	NA	NA	NA	88	92	NA
Chen et al. (2021) [33]	84	272	23	34	85	75	0.19	89	90	0.49
Shenhar et al. (2021) [34]	24	37	NA	NA	85	84	0.71	89	92	0.96
Shen et al. (2022) [35]	23	42	30	33	95	95	0.99	NA	NA	NA

EA: endoscopic ablation; RNU: radical nephroureterectomy; OS: overall survival; CSS: cancer-specific survival; NA: not available.

**Table 2 cancers-15-04495-t002:** Oncological outcomes of studies comparing endoscopic ablation vs. RNU for UTUC, stratified by tumor grade.

Study (yr) [Ref]	Patients (*n*)		Garde	5 yr OS (%)			5 yr CSS (%)			5 yr MFS (%)		
	EA	RNU		EA	RNU	*p* Value	EA	RNU	*p* Value	EA	RNU	*p* Value
Rouprêt et al. (2006) [36]	43	54	Low	NA	NA	NA	81	84	0.89	NA	NA	NA
Lucas et al. (2008) [28]	39	77	Low	75	66	0.28	86	87	0.91	NA	NA	NA
			High	45	72	0.08	69	75	0.53			
Gadzinski et al. (2010) [37]	34	62	Low	75	72	0.30	100	89	0.63	94	88	0.25
			High	25	48	0.62	86	72	0.94	86	64	0.79
Cutress et al. (2012) [29]	59	70	G1	75	86	0.62	100	100	0.65	NA	NA	NA
			G2	56	73	0.08	62	92	0.03			
			G3	33	75	0.001	83	89	0.26			
Grasso et al. (2012) [39]	80	80	Low	74	88	NA	87	93	NA	84	95	NA
			High	0	68	NA	0	78	NA	0	61	NA

EA: endoscopic ablation; RNU: radical nephroureterectomy; OS: overall survival; CSS: cancer-specific survival; MFS: metastasis-free survival; NA: not available.

**Table 3 cancers-15-04495-t003:** Renal function changes in contemporary studies comparing endoscopic ablation vs. RNU for UTUC.

Study (yr) [Ref]	Patients (*n*)		Renal Function			
	EA	RNU	Variable	EA	RNU	*p* Value
Fajkovic et al. (2013) [30]	20	178	Preoperative Cr (mg%) Postoperative Cr (mg%)	1.46 ± 0.52 1.3 ± 0.47	1.53 ± 1.2 1.64 ± 0.79	0.82 0.048
Hoffman et al. (2014) [42]	25	22	Preoperative eGFR Postoperative eGFR	66 62	68 58	>0.05 >0.05
Wen et al. (2018) [43]	32	107	Cr level POD1 (umol/L)	89 ± 7.5	123 ± 9.4	<0.01
Chen et al. (2021) [33]	84	272	Preoperative Cr (mg/dL) Postoperative Cr (1 mo) Postoperative Cr (final) ESRD	2.1 ± 1.9 3.57 ± 10.5 3.34 ± 3.01 29%	1.33 ± 2.82 1.61 ± 2.49 1.80 ± 2.73 27%	0.90 0.38 0.74 0.31
Shenhar et al. (2022) [34]	24	37	eGFR (mL/min/1.73 m^2^) ^#^ CKD (GFR < 60) Severe CKD (GFR < 30)	58.7 ± 21.5 45% 9%	49.2 ± 22.1 70% 16%	0.12 0.59 0.44

^#^ All variables were measured at the end of follow-up (median 5 years). Cr: creatinine; GFR: glomerular filtration rate; CKD: chronic kidney disease; ESRD: end-stage renal disease; POD: postop day.

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
