# Peer review of "Modern Kidney-Sparing Management of Upper Tract Urothelial Carcinoma"

_cancers, 2023, doi:10.3390/cancers15184495_

Round 1
Reviewer 1 Report
This review manuscript provides a comprehensive overview of the current treatment options available for managing upper tract urothelial carcinoma (UTUC) within the urinary tract. The manuscript effectively outlines the merits and drawbacks associated with three main treatment approaches: radical nephroureterectomy, endoscopic ablation, and segmental ureteral resection. By consolidating these crucial concepts, the manuscript proves to be a valuable resource for its readers.
While the Introduction section lays a succinct foundation, there is room for expansion to furnish additional insights into various aspects of UTUC. Expanding this section to encompass details such as prevalence, survival rates, prognosis, common etiological factors, and the genetic underpinnings of kidney cancer would enrich the reader's understanding of the subject matter.
In light of enhancing the manuscript's utility, the Conclusion section could be further developed. By incorporating information about emerging and innovative treatment modalities for kidney cancer, the manuscript would cater to the readers' curiosity and provide them with a more holistic view of the field's progress.
Overall, the manuscript adeptly compiles and presents a wealth of information regarding UTUC treatment options. Expanding the Introduction and Conclusion sections as suggested would undoubtedly contribute to the manuscript's comprehensiveness and value to its readers.
Author Response
Thank you so much for your comments. The introduction has been updated per your valuable comment (page 2, lines 40-46). Also, more details about the innovative treatments of UTUC have been added to the conclusion (page 8, lines 398-401).
Reviewer 2 Report
In this paper, the authors review modern techniques and outcomes of kidney-sparing surgeries (KSS) in patients with upper tract urothelial carcinoma (UTUC).
To this end, a comprehensive literature search on the study topic was conducted before April 30, 2023, using electronic databases, including PubMed, MEDLINE, and EMBASE. A narrative overview of the literature was then provided based on the extracted data and a qualitative synthesis of the findings.
The authors report that KSS is recommended for low- as well as select high-risk UTUCs who are not eligible for radical treatments. Endoscopic ablation is a KSS option that is associated with similar oncological outcomes compared to radical treatments while preserving renal function in well-selected patients. The other option in this setting is distal ureterectomy, which has the advantage of providing a definitive pathological stage and grade. Data from retrospective studies support the superiority of this approach over radical treatment with similar oncological outcomes, albeit in select cases. Novel chemoablation agents have also been studied in the past few years, of which mitomycin gel has received FDA approval for use in low-risk UTUCs.
The authors conclude that KSSs are acceptable approaches for patients with low- and select high-risk UTUCs, which preserve renal function without compromising the oncological outcomes.
Overall, this is a timely and interesting paper. I commend the authors on this nice paper.
Author Response
We sincerely thank the reviewer for the comments.
Reviewer 3 Report
This review summarizes current literature on nephron-sparing for upper urothelial carcinoma.
Well written and clear.
Despite the low quality of studies, this review includes the best literature up to now.
Please consider adding
DOI: 10.3389/fonc.2022.985177
DOI: 10.3390/jcm12154907
DOI: 10.1016/j.euf.2022.11.014
if appropiate
Author Response
Thank you for your comments. We added 2 suggested papers (DOI: 10.3389/fonc.2022.985177 and DOI: 10.3390/jcm12154907) to the manuscript (lines 116-118 [Ref 15] and lines 129-131 [Ref 21]).